# Amniocentesis—When It Is Clear That It Is Not Clear

**DOI:** 10.3390/jcm12020454

**Published:** 2023-01-06

**Authors:** Razvan Ciortea, Andrei Mihai Malutan, Carmen Elena Bucuri, Costin Berceanu, Maria Patricia Rada, Cristina Mihaela Ormindean, Dan Mihu

**Affiliations:** 12nd Department of Obstetrics and Gynaecology, “Iuliu Hațieganu” University of Medicine and Pharmacy, 400012 Cluj-Napoca, Romania; 2Department of Obstetrics and Gynecology, Emergency University Hospital Craiova, The University of Medicine and Pharmacy of Craiova, 200642 Craiova, Romania

**Keywords:** amniocentesis, contamination, technique, needle gauge, doppler, pregnancy, risk

## Abstract

A fetus identified to be at risk for chromosomal abnormalities may benefit from identification of genetic defects through amniocentesis. Although the risks associated with amniocentesis are considered to be minimal, being an invasive procedure it is not completely without complications. Background and Objectives: The current study aims to identify correlations between blood contamination of samples collected during amniocentesis and certain factors dependent on the instruments used (thickness of the needle used to aspirate the fluid), the location of the placenta, and uterine vascularity (more pronounced in multiparous patients). Materials and Methods: The study included 190 patients in the second trimester of pregnancy who met one of the criteria for invasive prenatal diagnosis (age over 35 years, high risk in first trimester screening, history of pregnancies with genetic abnormalities, etc.). The amniotic fluid samples collected from these patients were analyzed in terms of blood contamination of the amniotic fluid aspirated with maternal cells Results: Of the patients in whom the procedure was performed using 21 G size needles, 16 samples (13.33% of the total) were contaminated. None of the samples collected from patients where a 20 G needle was used were contaminated. There was a statistically significant association between the lack of contamination and the use of Doppler ultrasound in multiparous patients with anterior placenta in whom a 21-gauge needle was used for amniocentesis. Conclusions: There is an increased rate of sample contamination (statistically significant) when using 21 G needle sizes and a significant difference in contamination between primiparous and multiparous patients, with contamination being more frequent in multiparous patients. The use of Doppler ultrasonography may benefit the procedure, as the contamination rate was significantly reduced when used during amniocentesis.

## 1. Introduction

Amniocentesis is an invasive procedure performed primarily in the second trimester of pregnancy to establish a prenatal diagnosis [1,2]. It was first performed in 1967, and by the mid-1970s it was accepted as a tool used for prenatal diagnosis [3]. It involves obtaining fetal cells from the amniotic fluid by ultrasonographically guided puncture of the amniotic sac. A fetal ultrasound is performed prior to amniocentesis to confirm fetal viability, gestational age, number of fetuses, placental location, amniotic fluid volume, fetal anatomical survey, uterine cavity abnormalities, or presence of fibroids. The procedure is performed to identify any fetal chromosomal anomalies [2,4]. A fetus identified to be at risk for chromosomal abnormalities may benefit from identification of genetic defects through amniocentesis. On the one hand this helps the family in making an informed decision on whether or not to continue the course of pregnancy, preparation for delivery, and neonatal prognosis, and on the other hand it helps the physician in making a diagnosis [1,5].

The indications for amniocentesis may vary depending on the geographical region, medical centre, personal, or family history of the patient or risk factors identified for each individual patient. Patients over 35 years of age, those with a family history of genetic abnormalities, patients with a history of children with different genetic syndromes, pregnancies in which abnormalities are ultrasonographically identified, and pregnancies with positive screening for abnormalities may benefit from this diagnostic procedure [1,2,6].

Since 2015, the number of pregnant patients undergoing this type of invasive investigation has started to progressively and steadily decrease due to the development of non-invasive methods of prenatal testing (NIPT) that involve the identification of fetal DNA from maternal blood. Based on sequencing of cell-free fetal DNA (cff DNA) from maternal plasma (NIPT), professional societies have issued their present positions on non-invasive testing regarding Down syndrome (trisomy 21), in addition to other autosomal aneuploidies (trisomy 18 and 13 [7,8,9,10]). All existing statements emphasize that NIPT should not be offered as a diagnostic test for fetal aneuploidy. Many declarations also insist that there is lacking evidence for NIPT to be used as a screening test in a wide population, even though of late there have been some studies that validate good performance in women at average risk. DNA sequenced represents a combination of maternal and fetal cell-free DNA, with the latter actually originating from the placenta, thus rendering NIPT for common autosomal aneuploidies less than fully accurate [11]. A positive result (beckoning a supposed aneuploidy) may be created by factors other than an aneuploid fetal karyotype, therefore including placental mosaicism, a vanishing twin or a maternal tumor; false alarms are foreseeable [12].

NIPT is a much more accurate examination for common autosomal aneuploidies than cFTS. Nevertheless, a positive NIPT result should not be looked upon as a concluding diagnosis, as the placenta contributes to the cff DNA. Women should consequently be advised to have a positive result confirmed through invasive testing, if possible, by amniocentesis [13].

Therefore, amniocentesis and chorionic villus sampling remain the preferred methods of diagnosis [5,14]. Although the risks associated with amniocentesis are considered to be minimal, being an invasive procedure, it is not completely without complications. These include loss of amniotic fluid, which can occur both during and after the test. Most often this disappears within a week, but in rare cases it may continue throughout pregnancy increasing the maternal and fetal risk for infection, fetal compromise (cardiac compression) and preterm birth [2,4,5]. Another complication associated with amniocentesis, described in the literature, is fetal injury during maneuvering, which is why amniotic sac puncture is performed under ultrasonographic guidance. One of the most feared complications is pregnancy loss, but the rate of pregnancy loss after amniocentesis is less than 1% [2,3,5].

A recent study shows that the complication rate associated with amniocentesis may be related to the thickness of the needle used to perform the procedure, the number of punctures performed, the puncture performed transplacentally and, last but not least, the experience of the operator. Literature data show a need for at least 30 procedures performed annually to maintain manuality in performing the diagnostic maneuver, as well as to minimise risks related to the experience of the performer [2,5].

Contamination of amniotic fluid collected during amniocentesis with maternal cells has been identified to be responsible for diagnostic errors as early as 1976 [15]. The maternal cells are thought to be artificially introduced into the amniotic fluid sample, as a result of placental bleeding during amniocentesis [16], but they also can be maternal cells from all tissues that are punctured during the procedure. In 1983 Benn et al. [17] reported a decrease in the contamination rate of samples by removing the first millilitres of amniotic fluid collected, with a 2.5-fold lower contamination rate [15,18].

The current study aims to identify correlations between blood contamination of samples collected during amniocentesis and certain factors dependent on the instruments used (thickness of the needle used to aspirate the fluid), the location of the placenta, and uterine vascularity (more pronounced in multiparous patients).

## 2. Materials and Methods

The present study is a prospective cohort study, carried out in the period 2016–2021 in the Obstetrics-Gynecology Clinic “Dominic Stanca” Cluj Napoca, Romania. The study included 190 patients in the second trimester of pregnancy (between 16 and 19 weeks of gestation) who met one of the criteria for invasive prenatal diagnosis (age over 35 years, high risk in prenatal screening, history of pregnancies with genetic abnormalities, family history, pregnancies with ultrasonographic anomalies). Patients where the procedure was performed using needles with other sizes (18 or 22 G), patients with increased risk of amniotic fluid contamination (uterine fibroid located in the anterior uterine wall, patients with haematological pathologies, and those with changes in coagulation parameters), and also patients who required multiple punctures for amniotic fluid collection, or the ones in which the procedure needed to be repeated, were not included in the study Figure 1.

This prospective cohort research was conducted after obtaining the approval of the Ethics Commission of the University of Medicine and Pharmacy “Iuliu Hațieganu” Cluj Napoca (73/22.03.2016) and the informed consent of the patients included in the study group.

The amniotic fluid samples collected from these patients were analyzed in terms of blood contamination of the amniotic fluid aspirated with maternal cells and an attempt was made to make associations according to the following parameters: thickness of the needle used to perform amniocentesis (20–21 G), location of the placenta (at the level of the anterior/posterior uterine wall), uterine vascularization (parity-primipara/multipara).

The analysis was performed on patients who met one of the three criteria, two of the three criteria, or all three criteria. For the latter, additionally data were also analysed according to whether or not Doppler ultrasound was performed.

The technique of performing amniocentesis involves preparation of the patient’s tegument and its disinfection with antiseptic solution, followed by puncture of the tegument, under ultrasonographic guidance, Doppler mode was employed only during the insertion of the needle in order to choose a less vascularized area of the uterus or placenta, using needles with sizes between 20–21 G until the amniotic sac is punctured. Once penetrated into the cavity, 15–30 mL of amniotic fluid is aspirated, none of the aspirated fluid is disposed. The collected sample is then sent to the laboratory for fetal karyotype analysis. For establishing the contamination of the collected samples the test method used involves the numerical analysis of short repetitive sequences specific to each individual, called STRs, through QF-PCR (Quantitative Fluorescence—PCR) technique, which is based on multiplex-PCR amplification using fluorescent labelling. DNA extraction was performed from the processed sample, followed by PCR reaction using the Devyser Compact v3 kit. PCR products were migrated through capillary electrophoresis using the ABI3500 Genetic Analyzer. Interpretation was performed using GeneMapper analysis software. Fetal and maternal STR markers were analysed in parallel and contamination of the amniotic fluid sample with maternal cells was analyzed. Prenatal sampling procedures carry a risk of including maternal cells alongside the envisioned fetal sample. This risk has been empirically calculated as approximately 0.5% in amniotic fluid sampling, 1–2% in chorionic villi sampling, and can possibly be higher, reliant on the method used for invasive sampling. The markers present in the putative fetal DNA sample (derived from cultured or uncultured cells from amniotic fluid, cord blood, chorionic villi, or products of conception) are compared to those from a maternal DNA sample (from a maternal blood specimen). By inspecting these samples in parallel and associating the relative ratios of alleles at each individual marker, it is possible to estimate the presence and level of contaminating maternal cells in the fetal sample. The finding of MCC does not automatically dictate that a repeat invasive sampling procedure be recommended or performed. The mere presence of MCC does not impede diagnostic testing, while the test employed is robust to the level of contamination observed. Common cutoffs include 10% for SNP oligonucleotide analysis (SOMA) and 15% for direct sequencing-based tests [19]. These common cutoffs are the ones we used in our assessments.

We have excluded visual analyzing of samples because data from the literature show that even in macroscopically clear samples maternal cells can be identified. Amniotic fluid samples contamination with maternal cells, depending on the proportion of maternal cells identified, can reduce the accuracy of the result, and may also lead to the impossibility of interpreting the sample.

For amniocentesis, 20 G needles were used in 70 patients who met the inclusion criteria between 2016 and 2018 and 21 G needles in 120 patients who met the inclusion criteria for our study between 2019 and 2021. Amniocentesis procedures were performed either with a 120 mm 20-gauge (G) needle specific for amniocentesis and chorionic villous sampling (Egemen International Amniocentesis Needle, İzmir, Turkey), or with a 120 mm 21 G amniocentesis needle (Wallace Amniocentesis Needles, Cooper Surgical, Trumbull, CT, USA). Both needles are provided with an adjustable stopper to determine insertion depth, and with a female luer lock connector to collect the amniotic fluid through a syringe.

Statistical processing of the data was performed using the χ^2^ (chi-square) statistical test. The significance threshold for the statistical test used was α = 0.05 (5%) and risks were also calculated (RE—risk in the exposed, RN—risk in the unexposed and RR—relative risk or risk ratio).

## 3. Results

From May 2016 to July 2021, more than 200 diagnostic amniocenteses were performed in our institution, and of these a significant sample, 190 patients, consented for enrollment in the study.

The location of the placenta was in the anterior uterine wall in 55% of the study participants, the rest (45% of the patients) had the placenta located in the posterior uterine wall. In terms of uterine vascularity during pregnancy, a variable directly influenced by the parity of the patients, 38% of those included in the study group were primiparous, and the remaining 62% of patients were multiparous.

Samples were analyzed according to the fulfillment of one to all three criteria (thickness of the needle used to perform amniocentesis, location of the placenta, uterine vascularization).

### 3.1. One of Three Criteria Used

Of the total 190 patients included in the study, in 120 (63.16%) amniocentesis was performed using a 21 G needle and in 70 (36.84%) the procedure was performed using a 20 G needle. Of the patients in whom the procedure was performed using 21 G size needles, 16 samples (13.33% of the total) were contaminated. None of the samples collected from patients where a 20 G needle was used were contaminated. There was a statistically significant association between the use of a 21 G needle for amniotic sac puncture and contamination of collected samples (*p* = 0.0035) Figure 2.

In the patients included in the study, the contamination of the samples was analyzed taking into account the location of the placenta in the anterior or posterior uterine wall; thus from the group of 190 patients, in 105 of them (55.26%) the placenta was located anteriorly, out of which when performing amniocentesis in a group of 34 patients the sample was collected by punctioning across the placental tissue, and in 85 (44.74%) the placenta was located posteriorly. Of the patients with anterior placenta, in 13 patients (6.84% of the total) the sample collected was contaminated. Amongst patients with posterior placenta, only three patients (1.58% of the total) had contaminated samples. No statistically significant association could be observed between placental position and contamination of samples collected for prenatal diagnosis (*p* = 0.0546, RE = 12.38, RN = 3.53, RR = 3.51).

The last criterion for data analysis considered was the number of previous pregnancies of the patients, thus out of the group of 190 patients, 116 of them (61.05%) were multiparous and 74 (38.95%) were primiparous. Of the multiparous patients, in 14 patients (7.37% of the total) the samples collected showed contamination, and of the primiparous patients, in 2 of them (1.05% of the total) the samples were classified as contaminated. A statistically significant association was observed for sample contamination in multiparous versus primiparous patients (*p* = 0.0456) Figure 3.

### 3.2. Two of the Three Criteria Used

A total of 105 patients had placenta located in the anterior uterine wall; in 80 of them (76.19%) amniocentesis was performed using 21 G needles, and in 27 patients the sample was collected by punctioning across the placenta. In the other study participants, a number of 25 patients (23.81%), out of which in 7 cases the sample was collected through placental punctioning, the operation was performed with 20 G needles. Of the patients in whom 21 G needles were used for amniocentesis, in 13 patients (12.38% of all patients with anteriour placenta) the samples were contaminated. In none of the patients in whom amniotic sac puncture was performed using 20 G needles was there contamination of the samples. There was a statistically significant association between the use of a 21 G needle to perform amniocentesis and contamination of samples obtained from patients with previous placenta (*p* = 0.0226) Figure 4.

The correlation between patient parity and the size of the needle used to perform the maneuver was also analyzed. Out of the total 116 multiparous patients, in 74 of them (63.79%) amniocentesis was performed using a 21 G needle and in 42 of them (36.21%) a 20 G needle was used for the maneuver. Among the patients in whom 21 G needle was used for amniotic fluid collection, samples collected from 14 patients (12.07% of all multiparous patients) were contaminated, but there were no contaminated samples in patients in whom 20 G needle was used. There was a statistically significant association between the use of a 21 G needle for amniocentesis in multiparous patients and contamination of amniotic fluid samples (*p* = 0.007) Figure 5.

A correlation was also sought between the use of a 21 G needle and multiparity for amniocentesis. Thus, out of the total of 120 patients in whom a 21 G needle was used for amniocentesis, 74 (61.67%) were multiparous and 46 (38.33%) were primiparous. Of the multiparous patients, 14 patients (11.67% of the total number of patients who used a 21-gauge needle for amniocentesis) had contaminated samples, and of the primiparous patients, only two samples had maternal cell contamination (1.67% of the total number of patients who used a 21-gauge needle for amniocentesis). A statistically significant association was observed between the use of a 21 G needle for amniocentesis and contamination of samples collected from multiparous patients (*p* = 0.0448) Figure 6.

In patients who met two of the three criteria established, correlations were also sought between anterior placental location and patient parity, placental location at the posterior uterine wall and multiparity or primiparity, multiparity and anterior or posterior placental location, and primiparity and anterior or posterior placental location, posterior location of the placenta and size of needle used for amniocentesis (20 G or 21 G), use of 21 G needle and location of the placenta in the anterior or posterior uterine wall, and primiparity of the patient and thickness of needle used (20 G or 21 G), with no statistically significant associations between these criteria.

### 3.3. All of the Three Criteria Used

The presence of a statistically significant association between the use of a 21 G needle in multiparous patients and the location of the placenta in the anterior uterine wall compared to the use of 20 G needles in primiparous patients and a location of the placenta in the posterior uterine wall was analyzed.

Out of the total of 49 multiparous patients with anterior placenta in whom amniotic sac puncture was performed using 21 G needles, in 23 patients the punction was realised across the placenta, and in 11 of them (22.45% of them) maternal cells were identified present in the collected samples. None of the samples collected from primiparous patients in whom the placenta was identified in the posterior uterine wall and a 20 G needle was used to puncture the amniotic sac showed contamination.

There was a statistically significant association between the use of a 21 G needle for amniocentesis in multiparous patients with anteriorly located placenta and contamination of amniotic puncture samples (*p* = 0.0454) Figure 7.

For patients who met all three criteria, we also looked for correlations between the size of the needle used (20 G or 21 G), the anterior location of the placenta in multiparous patients, the size of the needle used (20 G or 21 G), the anterior location of the placenta in primiparous patients, the use of 21 G needles, the anterior location of the placenta and the parity of the patients, and the use of 21 G needles. Posterior location of the placenta and parity of the patients and correlations were also sought between needle size used for amniocentesis (21 G), parity (primiparous and multiparous) and anterior or posterior location of the placenta, but no statistically significant associations could be found to correlate these criteria with contamination of amniocentesis samples (*p* > 0.05). Using these conditions it was however observed that 21 G needles were used for all samples collected that showed contamination.

Further research was carried out to find ways in which the amniocentesis procedure could be improved to avoid contamination. Thus, the use of Doppler ultrasound during the maneuver was attempted and the data were then analyzed using the same statistical tests and reference intervals.

We looked for a difference between performing or not performing Doppler ultrasound and contamination of samples obtained from multiparous patients with placenta located in the anterior uterine wall and in whom 21 G needles were used for amniocentesis.

Of the total 49 multiparous patients with anterior placenta in whom 21 G needles were used for amniotic fluid collection, in 39 of them (79.59%) the collection was performed under Doppler ultrasonographic guidance and in 10 of them (20.41%) using only grayscale (2 D) ultrasonography. Of the patients in whom Doppler ultrasonography was used, in two patients (4.08% of the total multiparous patients with anterior placenta in whom 21 needle was used for amniocentesis) maternal cells were identified in the samples collected. Of the patients in whom Doppler ultrasound was not used, nine patients (18.37% of all multiparous patients with previous placenta in whom a 21-gauge needle was used for amniocentesis) showed contamination in the samples collected.

There was a statistically significant association between the lack of contamination and the use of Doppler ultrasound in multiparous patients with anterior placenta in whom a 21-gauge needle was used for amniocentesis (*p* < 0.0001) Figure 8.

This method has also been tested to improve amniocentesis performance using 21 G needles in primiparous patients with placenta located at the anterior uterine wall was also tested. Thus, out of a total of 31 primiparous patients with anterior placenta in whom 21 G needles were used for amniocentesis, Doppler ultrasound was performed in 27 of them (87.10%) and not in 4 of them (12.90%). None of the patients in whom Doppler ultrasound was used had contaminated samples, and of the patients in whom Doppler ultrasound was not used, two patients (6.45% of all primiparous patients with previous placenta in whom 21 G needle was used for amniocentesis) had contaminated samples.

There was a statistically significant association between lack of contamination and the use of Doppler ultrasound in primiparous patients with previous placenta in whom a 21-gauge needle was used for amniocentesis (*p* = 0.0146) Figure 9.

Comparative analysis of the group of patients with placenta located in the anterior uterine wall who underwent amniocentesis using 21 G size needles and Doppler ultrasound to perform amniocentesis showed that out of the total 66 patients, 39 of them (59.09%) were multiparous and 27 (40.91%) were primiparous. Among the multiparous patients, two patients (3.03% of those in whom previous placental location was detected in which 21 G needle was used for amniocentesis and Doppler ultrasound was used) had samples contaminated with maternal cells. None of the samples collected from primiparous patients were contaminated. No statistical significance was found (*p* = 0.6).

The reduction in contamination of samples collected during amniocentesis using 21 G needles and Doppler ultrasound was also analyzed for posterior placental location in both primiparous and multiparous patients.

Of the total 25 multiparous patients with posterior placenta in whom 21 G needles were used for amniocentesis, 18 of them (72.00%) benefited from the use of Doppler ultrasonography and in seven (28.00%) it was not used. None of the patients in whom Doppler ultrasound was performed had contaminated samples. Of the patients in whom Doppler ultrasound was not performed, in three patients (12.00% of the total of multiparous patients with posterior placenta in whom 21 needles were used for amniocentesis) the samples showed maternal cell contamination.

There was a statistically significant association between the lack of contamination of amniotic fluid samples collected and the use of Doppler ultrasound in multiparous patients with posterior placenta in whom a 21 G needle was used for amniocentesis (*p* = 0.0152) Figure 10.

The existence of statistically significant associations between the use of 21 G needles, placenta with posterior uterine wall insertion, primiparous patients, the existence of a difference between primiparous and multiparous patients, and the non-use of Doppler ultrasound during the maneuver was also investigated, without identifying the existence of statistically significant correlations.

Data and results from the present study are listed in Table 1.

## 4. Discussion

Amniocentesis is currently the most widely used antenatal diagnostic method. Over time, the way of performing the maneuver has been improved in order to reduce the risks associated with amniotic sac puncture, but these risks, although considered minimal, must be known by the couple undergoing such a procedure [3,6]. Amniocentesis is a minor surgical procedure, usually performed in the second trimester of pregnancy to ensure that an optimal number of fetal cells are extracted from the amniotic fluid [1,15]. Amniocentesis is associated with higher rates of successful, clear taps, and lower rates of bloody taps (reduced from 2.4% to 0.8%) when performed under direct ultrasound control with continuous needle tip visualization. Best practice is that ultrasound scanning during the procedure be performed by the person inserting the needle [18]. When we also utilize the Doppler function, we enhance the safety of the maneuver, taking into account the added possibility of visualization of increased flow through some parts of the placenta, which can then be avoided. This is one of the major advantages of the Doppler feature. Additional advantage consists of avoiding a major complication, specifically puncturing the umbilical cord, which is certainly better visualized through Doppler examination, precisely useful in oligohydramnios situations where the cord has less freedom of motion. One of the ways to evaluate the success of this procedure is by assessing the postprocedural abortion rate which reaches 0.6% for ultrasound-guided maneuvers. Another criterion used to assess the success of the procedure is the quality of the specimen extracted, as contamination with maternal cells can lead to diagnostic errors [15,16].

Most studies in the literature have focused on how to perform amniocentesis in order to reduce the risks associated with the maneuver and very few studies have focused on the association between the size of needles used to perform amniotic sac puncture and the occurrence of complications. In this direction, the aim of our study was to identify the increased risk of specimen contamination taking into account both maternal factors (placental location, parity) and extrinsic factors (instruments used to perform the procedure, more specifically the size of needles used to perform amniocentesis). At the same time, a method to reduce the risk of contamination of the samples by using Doppler examination during the procedure was analyzed. Current research results show a significant association between maternal cell contamination of amniotic fluid samples collected during amniocentesis and the use of fine needles of smaller diameter (21 G). This finding is supported by two studies: the first one was published by Peter A. Benn and Lillian Y.F. Hsu [17] and identified a higher rate of contamination of samples obtained after puncture of the amniotic sac using needles larger than 20 G, and the second one was conducted by Apostolos P., Athanasiadis et al. [20] who demonstrate that the use of smaller needle diameters (22 G) causes greater trauma to tissues during amniocentesis, primarily through the longer duration of the amniotic fluid aspiration process. At the same time, the present study identified a higher risk of contamination associated with the use of small puncture needles (21 G), regardless of the patient’s parity (primiparous or multiparous). The use of larger gauge needles (20 G) shortens the time to perform the procedure, but is associated with a higher rate of amniotic fluid leakage postprocedurally [21]. This hypothesis is demonstrated through an experimental study design by Devlieger R et al. who tested amniotic fluid leakage in patients undergoing Caesarean section at term [22].

There are conflicting data in the literature regarding the increased risk of contamination associated with placental location at the uterine cavity walls. The findings of the study led by Hockstein et al. [23] show a lack of association between the risk of contamination of collected amniotic fluid samples and placental location in the uterine cavity. In contrast, research by Nub et al. [16] found a significantly higher rate of maternal cells in samples collected from patients with placenta located in the anterior uterine wall. This hypothesis is also supported by the studies performed by Pergamentet et al. [23] and Giorlandino C et al. [24]. They concluded that the frequency of contaminated samples was higher in three different situations: when the collection was performed transplacentally, when more punctures of the amniotic sac were required for collection, and when the maneuver was performed by a less experienced physician [25]. The present study shows a significant association between the amniotic fluid contamination and the anterior location of the placenta, thus an increased rate of contamination in association with transplacental amniotic fluid collection. The risk of contamination is further increased by two factors: when the placenta is located on the anterior wall and when the amniotic fluid extraction is performed using small (21 G) needles. Apostolos P. Athanasiadis et al. [20] and Uludag S [26] supported the hypothesis in which the use of 20 G needles for amniocentesis reduces intrauterine bleeding and therefore amniotic fluid contamination.

Another risk factor for samples contamination is the parity of patients, showing a significant association between multiparity and frequency of contamination.

In this research the usefulness of Doppler ultrasound during amniocentesis was investigated, demonstrating a reduction in contamination through its use. This hypothesis is in addition to Jennifer Weida et al. [15] and Benn et al. [17] studies that showed a significant 2.5-fold reduction in the frequency of contamination after removal of the first 1–2 mL of aspirated amniotic fluid.

One of the strengths of the present study is the large number of patients included in the study, with 190 patients undergoing invasive genetic testing. Also, the analysis of the contaminated samples with maternal cells taking into account all the criteria, considering the fact that the literature data analyses the factors separately, is another strength of the current research. Data from the present study show a significant reduction in the risk of contamination while using Doppler ultrasound for identifying uteroplacental circulation during amniocentesis, regardless of the size of the needle used to perform the puncture.

The present study has some limitations, among which we mention the impossibility to quantify the degree of contamination of the samples in order to assess the risk of false positive or negative results. Also, the existence of a correlation between the size of the needle used to perform the maneuver and the number of maternal cells identified in the sample analyzed was not possible.

## 5. Conclusions

There is an increased rate of sample contamination (statistically significant) when using 21 G needle sizes and a significant difference in contamination between primiparous and multiparous patients, with contamination being more frequent in multiparous patients. There was a significantly higher rate of contamination of samples associated with amniotic sac puncture using 21 G needles, as opposed to the contamination rate observed when larger gauge needles (20 G) were used. The contamination rate was significantly higher in multiparous patients when 21 G needles were used. In multiparous patients, contamination of collected samples is more frequent when 21 G needles are used for maneuvering.

A higher rate of contamination is observed in multiparous patients with a previously located placenta in whom 21 G needles were used to collect amniotic fluid for prenatal diagnosis.

The use of Doppler ultrasonography clearly benefits the procedure, as the contamination rate was significantly reduced when used during amniocentesis regardless of the factors favoring contamination present. Thus, it is our strong belief that only Doppler ultrasound should accompany amniocentesis procedures. This hypothesis should open opportunities for future research into the usefulness of Doppler studies during invasive diagnostic maneuvers.

## Figures and Tables

**Figure 1 jcm-12-00454-f001:**
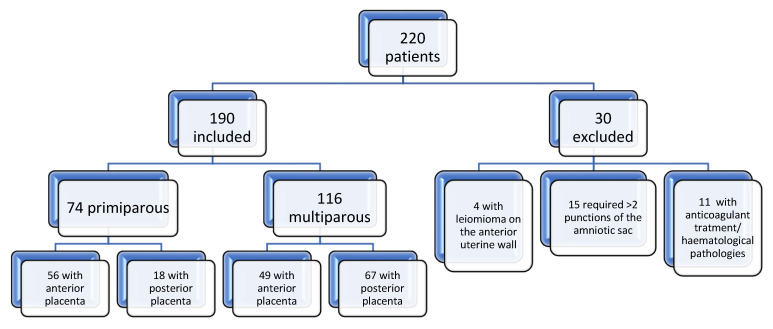
Inclusion and exclusion criteria for patients.

**Figure 2 jcm-12-00454-f002:**
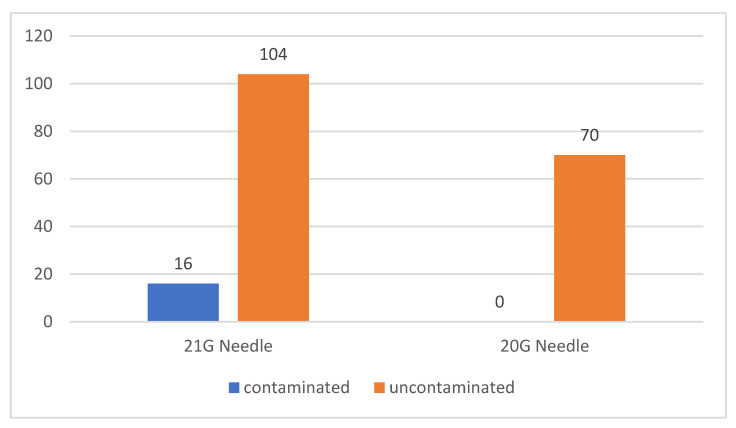
Number of contaminated versus uncontaminated samples according to the size of needle used for amniocentesis.

**Figure 3 jcm-12-00454-f003:**
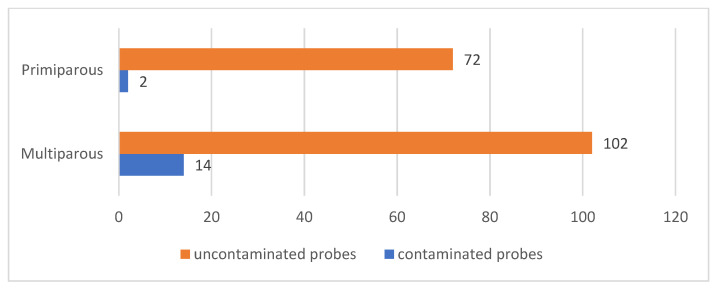
Number of contaminated versus uncontaminated samples according to vascularity during pregnancy (multiparous versus primiparous) regardless of needle gauge used.

**Figure 4 jcm-12-00454-f004:**
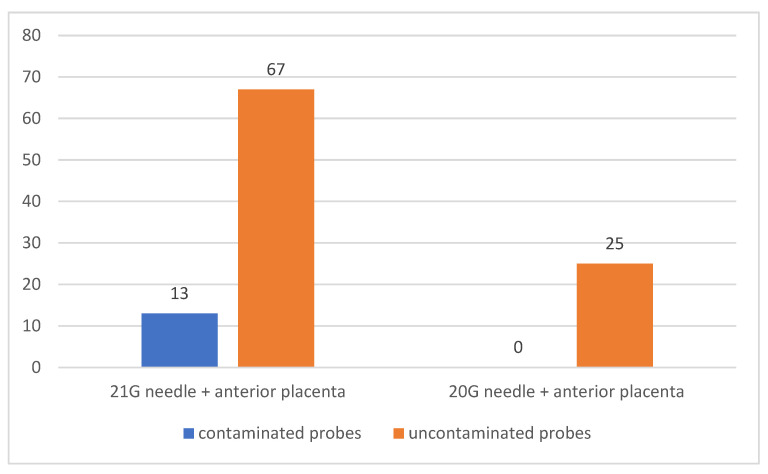
Sample contamination in patients with anteriorly located placenta correlated with the size of the needle used for sampling.

**Figure 5 jcm-12-00454-f005:**
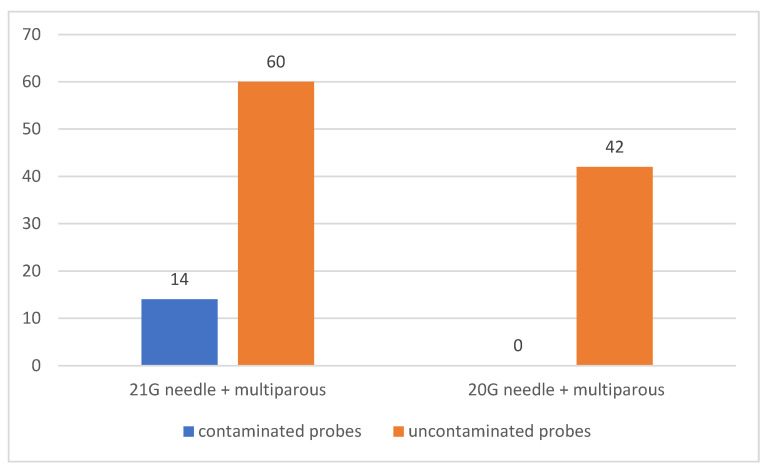
Contamination of samples based on multiparity and different gauge of needle used for collection.

**Figure 6 jcm-12-00454-f006:**
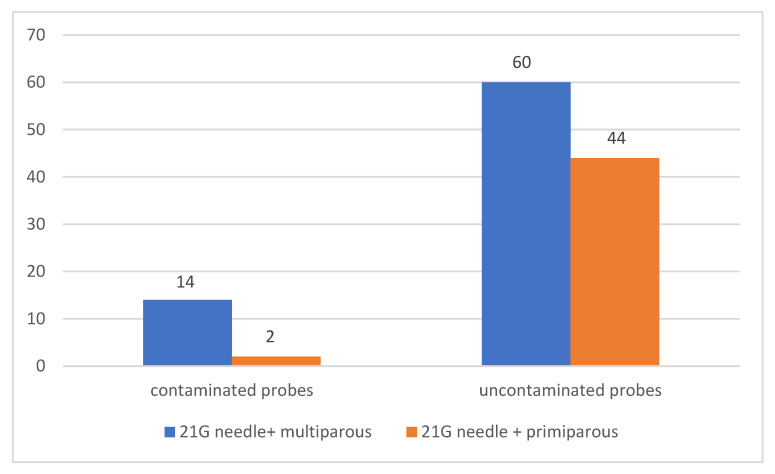
Sample contamination based on parity with 21 G needle.

**Figure 7 jcm-12-00454-f007:**
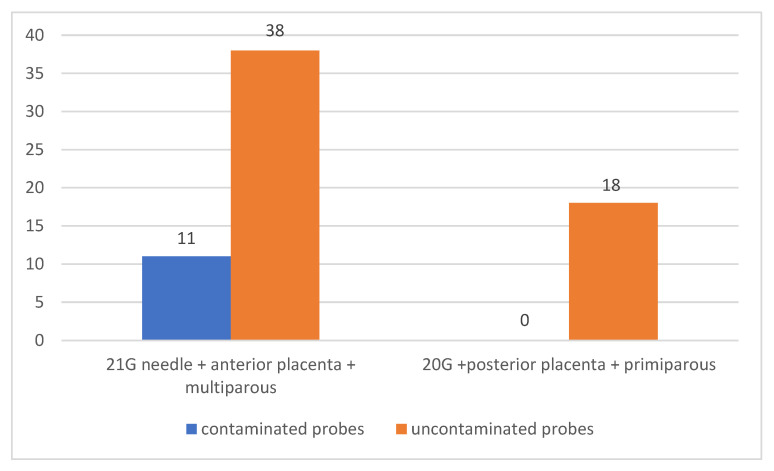
Contamination of samples according to the use of a 21 G needle in multiparous patients with placenta located at the anterior uterine wall compared to the use of 20 G needles in primiparous patients with placenta located at the posterior uterine wall.

**Figure 8 jcm-12-00454-f008:**
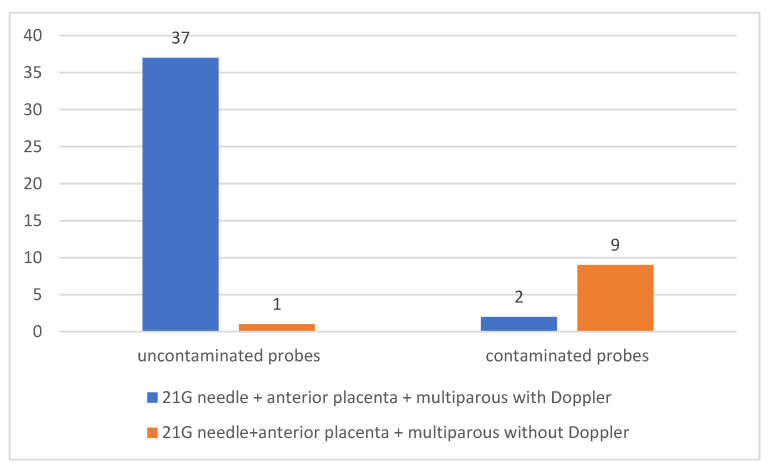
Use of Doppler ultrasound and reduction in contamination of samples collected using 21 G needles in multiparous patients with anteriorly located placenta.

**Figure 9 jcm-12-00454-f009:**
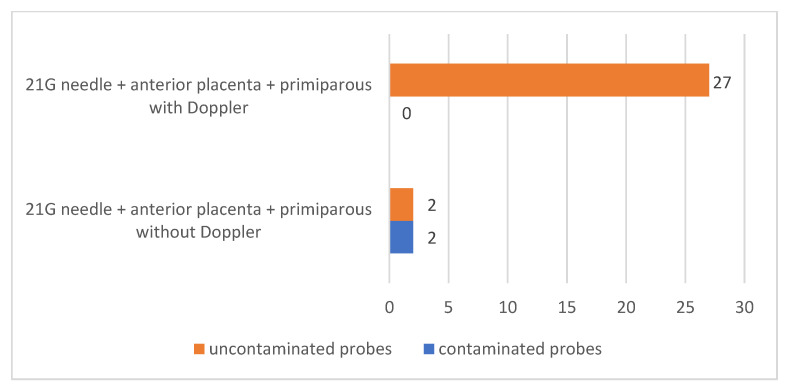
Association between lack of contamination of samples by performing amniocentesis using Doppler ultrasound.

**Figure 10 jcm-12-00454-f010:**
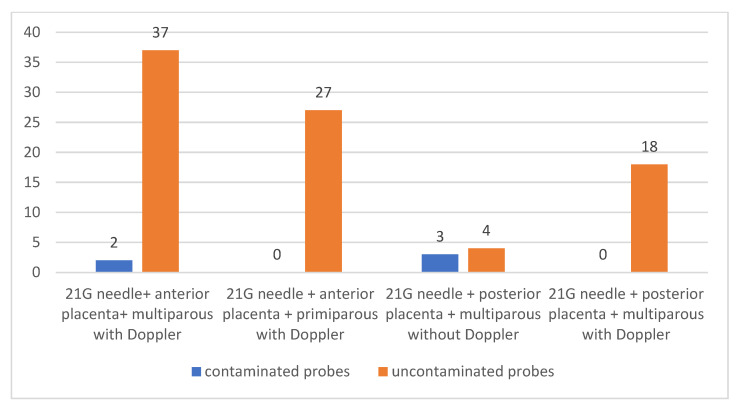
Reduction in contamination of amniocentesis samples using Doppler ultrasound.

**Table 1 jcm-12-00454-t001:** Study results. R_E_—Exposed risk, R_N_—unexposed risk, RR—relative risk.

Studied Criteria	Contaminated Probes	Uncontaminated Probes	*p* Value	R_E_		R_N_	RR
21 G Needle	16	104	0.0035	13.33		0	-
20 G Needle	0	70					
Multiparous	14	102	0.0456	12.07		2.703	4.466
Primiparous	2	72					
21 G needle + anterior placenta	13	67	0.0226	16.25		0	-
20 G needle + anterior placenta	0	25					
21 G needle + multiparous	14	60	0.007	18.92		0	-
20 G needle + multiparous	0	42					
21 G needle + multiparous	14	60	0.0448	18.92		4.348	4.351
21 G needle + primiparous	2	44					
21 G needle + anterior placenta + multiparous	11	38	0.0454	22.45		0	-
20 G needle + posterior placenta + primiparous	0	18					
21 G needle + anterior placenta + multiparous with Doppler	37	2	<0.0001	94.87		10	9.487
21 G needle + anterior placenta + multiparous without Doppler	1	9					
21 G needle + anterior placenta + primiparous with Doppler	27	0	0.0146	100		50	2
21 G needle + anterior placenta + primiparous without Doppler	2	2					
21 G needle + anterior placenta + multiparous with Doppler	2	37	0.6909	5.128		0	-
21 G needle + anterior placenta + primiparous with Doppler	0	27					
21 G needle + posterior placenta + multiparous without Doppler	3	4	0.0152	42.86		0	-
21 G needle + postrerior placenta + multiparous with Doppler	0	18					

## Data Availability

Not applicable.

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
