# Peer review of "Amniocentesis—When It Is Clear That It Is Not Clear"

_jcm, 2023, doi:10.3390/jcm12020454_

Round 1
Reviewer 1 Report
Journal of Clinical medicine (MDPI) jcm-2092763
Title : Amniocentesis-when it is clear that it is not clear.
Authors: Razvan Ciortea, Andrei Mihai Malutan, Carmen Elena Bucuri1, Costin Berceanu, Maria Patricia Rada1, Cristina Mihaela Ormindean and Dan Mihu.
Advice of the reviewer:
In terms of content, authors of this paper provide critical explanations on a way to improve the delicate technique of amniosynthesis by taking into account different parameters to avoid blood contamination of samples collected during amniocentesis: thickness (diameter?) of the needle used during the fluid aspiration, location of the placenta and vascularization (primi or multiparity), addition of ultrasonic guidance.
In terms of form, many things need to be reviewed and improved both in the chapters on materials and methods, results, figure legends and above all discussion.
Critical review and suggestions:
Introduction:
Blood contamination of samples : what is this mysterious contamination, what is it made of, blood cells, various fragments, membranes ?. Where does it come from? of the mother or of the placenta itself. It remains unclear.
Lack of information in the literature on the ultrasonic guidance (Doppler ultrasonography) and why it was chosen to be added during sampling. This method of ultrasound guidance seems to be classic now.
Numerotation of the bibliographic references does not respect the order of citation.
Material and methods: Explain the needle gauge chart and diameter of 21G and 20G needles. Indeed, a larger diameter favors a shorter sampling time. Explain the avantages of the ultrasonic guidance.
Results:
They are quite tedious to read: In each paragraph, it would be good to clarify each one with a small table.
At the end of this chapter, a short summary summarized on a table would be welcome.
Figure captions:
When looking at a figure it must enlighten the reader by giving him as much information as possible in a small space. In this work, no figure (10) is commented or explained correctly. Write or rewrite all figure legends.
Discussion: Verbose and with too long sentences and lots of repetitions. The advantage of ultrasound guidance is not discussed.
Final advice:
The reviewer rejects this paper for publication in JCM unless the authors make major revisions of the manuscript.
Reviewer 2 Report
Interesting idea that is becoming quite robust due to the large number of patients. Minor linguistic correction is required however.
Reviewer 3 Report
„Amniocentesis - when it is clear that it is not clear”
This is an interesting paper presenting the results of amniocentesis using needles of different diameters, which were carried out at two at two University Clinical Centres in Romania.
The work confirms previous reports of an increase in contamination of the material collected during amniocentesis when larger diameter puncture needles are used.
However, I have several comments:
1. In the Introduction section it is stated that:
"Since 2015, the number of pregnant patients undergoing this type of invasive
investigation has started to progressively and steadily decrease due to the development
of non-invasive methods of genetic testing (NIPT) that involve the identification of fetal
DNA from maternal blood. However, non-invasive tests are not definitive diagnostic
methods, which is why amniocentesis and chorionic villus sampling remain the preferred
methods of diagnosis” -
please specify in which indications amniocentesis can be abandoned based only on non-invasive methods of genetic testing (NIPT) and when amniocentesis is absolutely indicated.
2. In the Material and Methods section states :
''The study included 190 patients in the second trimester of pregnancy (between 16 and 19 weeks of
gestation) who met one of the criteria for invasive prenatal diagnosis (age over 35 years,
high risk in first trimester screening, history of pregnancies with genetic abnormalities,
etc.)" -
you cannot use the abbreviation etc - please provide all the criteria used by yours Centres.
3. In the Results section states: "The last criterion for data analysis considered was the number of previous pregnancies of the patients, thus out of the group of 190 patients, 116 of them (61.05%) were multiparous and 74 (38.95%) were primiparous.” -
can you explain why there was such a high percentage of multiparous patients? Were these patients after assisted insemination procedures?
4. In the Conclusions section :
it should be emphasised that amniocentesis should only be performed with the assistance of Doppler ultrasound, because only such a procedure reduces the risk of complications associated with the test and offers the chance to reduce the risk of contamination of the amniotic fluid samples collected during the test with maternal genetic material.

Reviewer 4 Report
In the current study the authors aimed to identify correlations between blood contamination of samples collected during amniocentesis and certain factors dependent on the instruments used (thickness of the needle used to aspirate the fluid), the location of the placenta, and uterine vascularity (more pronounced in multiparous patients).
Major revision
The major revision is about the quantification of the contamination. I did not understand how the authors evaluate the samples contamination. Did they recruit the maternal blood draw during the amniocentesis?
Minor revisions:
- In the introductionI suggest to replace the non-invasive methods of genetic testing with non-invasive prenatal testing (NIPT) and to add a sentence about the advantages and the limitations of NIPT
- In the methods I suggest to add a table with the demographic and the clinical data of pregnant women who performed the amniocentesis.
Results section
- The figure 1 is not clear to explain the % of contamination. I suggest to merge the figure 1 and the figure 2.
- In the figures add only the p-value, and not the p-value and p-value<0.05
- I suggest to merge the figures 3 and 4, although It's not clear how the authors add this figure, it means why the authors showed a correlation between parity and size if they already wrote about the increased contamination of 20G
- I suggest a different figure, with pimiparous versus multiparous with 21g needle
- I suggest to merge the figures 9 and 10.
Round 2
Reviewer 1 Report
I particularly appreciated the work of corrections and the addition of the Table 1 summarizing the results as well as the clearer discussion.
Reviewer 4 Report
The authors replied to all my comments.